# Validation of the performance of a point of care molecular test for leprosy: From a simplified DNA extraction protocol to a portable qPCR

Amanda Bertão-Santos[1], Larisse da Silva Dias[2], Marcelo Ribeiro-Alves[3], Roberta Olmo Pinheiro[2], Milton Ozório Moraes[2†], Fernanda Saloum de Neves Manta[2☯*], Alexandre Dias Tavares Costa [1☯*]

1 Laboratório de Ciências e Tecnologias Aplicadas à Saúde (LaCTAS), Instituto Carlos Chagas, Fundação Oswaldo Cruz (FIOCRUZ), Curitiba, Brazil, 2 Laboratório de Hanseníase, Instituto Oswaldo Cruz, Fundação Oswaldo Cruz (FIOCRUZ), Rio de Janeiro, Brazil, 3 Laboratório de Pesquisa Clínica em DST e AIDS, Instituto Nacional de Infectologia Evandro Chagas, Fundação Oswaldo Cruz (FIOCRUZ), Rio de Janeiro, Brazil

☯ These authors contributed equally to this work.
† Deceased.
* alexandre.costa@fiocruz.br, adtcosta@gmail.com; femanta@yahoo.com.br

**Data Availability Statement:** All relevant data are within the paper and its Supporting Information files.

## Abstract

The study aimed to optimize qPCR reactions using oligonucleotides from the first Brazilian molecular diagnostic kit for leprosy on a portable platform (Q3-Plus). In addition, we sought to develop a simplified protocol for DNA extraction that met point-of-care criteria. During optimization on the Q3-Plus, optical parameters, thresholds, and cutoffs for the *16S rRNA* and RLEP targets of *M. leprae* were established using synthetic DNA, purified DNA from *M. leprae*, and pre-characterized clinical samples. For the simplified extraction protocol, different lysis solutions were evaluated using chaotropic agents, and purification was carried out by transferring the lysed material to FTA cards. The complete protocol (simplified extraction + qPCR on the portable platform) was then evaluated with pre-characterized clinical skin biopsy samples and compared with standard equipment (QuantStudio-5). $LOD_{95\%}$ for the optimized reactions was 113.31 genome-equivalents/µL for *16S rRNA* and 17.70 genome-equivalents/µL for RLEP. Among the lysis solutions, the best-performing was composed of urea (2 M), which provided good dissolution of the skin fragment and a lower Ct value, indicating higher concentrations of DNA. The complete technological solution showed a sensitivity of 52% in reactions. Our results highlight the need for additional optimization to deal with paucibacillary samples, but also demonstrate the feasibility of the portable platform for the qPCR detection of *M. leprae* DNA in low infrastructure settings.

## Author summary

The diagnosis of tropical or neglected infectious diseases, such as leprosy, is highly dependent on techniques that are not very sensitive or very laborious, such as optical

**Funding:** This study was supported by grants from Carlos Chagas Filho Research Support Foundation of the State of Rio de Janeiro (Faperj, E-26/203.913/2022, www.faperj.br, to FSNM) Carlos Chagas Institute Research Stimulus Program (ICC 008 FIO 21 – SUB 22, www.icc.fiocruz.br, to ADTC) and by the National Fund for Health/Brazilian Ministry of Health (TED 69/2021, www.gov.br/saude/pt-br, to MOM). LSD is a CNPq fellowship holder, FSNM is a Faperj fellowship holder, and ADTC is a CNPq productivity fellow (level 2). The funders played no role in the study design, data collection and analysis, decision to publish, or preparation of the manuscript.

**Competing interests:** The authors have declared that no competing interests exist.

microscopy or culture. More sensitive and specific molecular tests are only available in urban centers, where there is a specialized workforce and adequate infrastructure. However, a large portion of the population does not have access to such medical services, either due to financial or distance issues. Thus, performing diagnostic tests at the point of care (POC) provides agility in diagnosis, avoiding the need for patients to travel to medical centers or transport biological samples to clinical analysis centers, ensuring that this population has access to diagnostic tests, even in remote areas or with poor infrastructure.

In this work, we optimize and validate, on a laboratory scale, simplified versions of all the procedures necessary to perform molecular tests, from collection and DNA extraction to the detection of genomic targets using portable qPCR equipment. The results presented here are the basis of a novel tool with potential to increase a leprosy screening test in remote areas, facilitating the active search of positive cases and monitoring by health authorities responsible for underserved populations.

## Introduction

Leprosy is a chronic and progressive infectious disease with worldwide distribution caused by *Mycobacterium leprae* or *M. lepromatosis* [1]. It presents tropism for peripheral nerves and skin, although other organs might also be affected. Progression of the disease might cause deformities with varying degrees of physical disability (GD). Given its signs and symptoms, leprosy can be manifested in a broad spectrum of clinical forms, some of which often lead to misdiagnosis with other dermatological, osteoarticular, or neurological conditions, and even other diseases [2,3].

Early diagnosis is essential for proper treatment and control of the disease's clinical progression and community transmission [4]. Due to its inability to grow *in vitro*, direct diagnostic techniques such as culture and isolation are not feasible [5]. Diagnosis is usually late or non-existent because it is mainly based on the patient's clinical and epidemiological information, guided by anamnesis and physical examination, which demands physician's expertise [6,7]. In the absence of a gold standard diagnostic test, complementary tests are often employed, such as histopathology and bacilloscopy of the slit-skin smear. These tests, however, exhibit variations in sensitivity accordingly the clinical form of the disease and rely on experience for contextualization and interpretation [2,6,7].

Molecular tests, particularly real-time polymerase chain reaction (qPCR), are sensitive and specific, contributing to the early identification of various pathologies [8,9,10]. Although molecular detection tests are available, the diagnosis of leprosy continues to rely primarily on clinical diagnosis [6]. This is due to the variations in clinical presentations, especially for paucibacillary cases, as the sensitivity of tests varies according to the bacillary load of the infection [11].

Recently, Brazilian health authorities (ANVISA) granted registration for the first national qPCR Nucleic Acid Testing kit (NAT Hans Kit, IBMP, Brazil), developed by the Oswaldo Cruz Foundation (Fiocruz/RJ) [12]. This kit specifically targets the genetic markers *16S rRNA* and RLEP *M. leprae*, demonstrating sensitivity and specificity of 91% and 100%, respectively. Additionally, it utilizes human *18S rRNA* as an internal control, ensuring proper DNA extraction. The incorporation of this molecular test into the routine of healthcare professionals engaged in active case finding for leprosy can overcome the intrinsic limitations of direct diagnostic methods such as bacilloscopy, histopathology, and indirect serology tests, thereby expanding the detection capacity, especially for the paucibacillary (PB) clinical form [11].

Tools that assist in population screening play a crucial role in active case finding and early diagnosis of leprosy. Screening tests exhibit higher sensitivity, meaning they can more accurately identify positive cases for the disease of interest. A positive result in a screening test should guide the patient toward further assessments that allow for an accurate diagnostic investigation, as is done in the case of leprosy [13]. However, the implementation of leprosy molecular detection tests in the field faces limitations. The requirement for thermolabile reagents and robust equipment hinders access to diagnosis in remote and low-infrastructure regions. Furthermore, the method requires prior DNA extraction, demanding investments in costly commercial kits and time-consuming to obtain the sample [14,15,16]. Regarding nucleic acid testing (NAT) based diagnosis, the primary challenges are related to the pre-analytical phase, including specimen collection and biological material extraction. The need for sensitive instruments, such as centrifuges, and high-cost reagents, as well as the proper disposal of the chemical residues generated in these steps, represents the primary limitations for their applicability in resource-limited settings [15,17].

The aim of the study was optimizing qPCR reactions using the NAT Hans kit (IBMP, Brazil) on the portable Q3-Plus instrument, while concurrently developing a simplified DNA extraction protocol for skin samples, aiming the detection of *M. leprae* DNA. The resulting prototype enables the implementation of a leprosy screening test in remote areas, facilitating the active search of positive cases and monitoring by health authorities responsible for underserved populations.

## Materials and methods

### Ethics statement

The present research project was approved by the Ethics Committee of the Oswaldo Cruz Institute (CAAE: 52565521.2.0000.5248, number: 5.131.588 approved on November 26, 2021). Suspected leprosy patients attending the Souza Araújo Out-Patient Unit (ASA), a leprosy reference center from Brazilian Ministry of Health at Oswaldo Cruz Institute–Fiocruz–RJ–Brazil, provided written consent to participate in the project. In the case of minors, formal written consent was obtained from the patient's guardian. The selection of samples was carried out according to the occurrence of cases attended at the clinic.

### Synthetic DNA

Synthetic double-stranded DNA (gBlock, IDT, USA) containing the sequences of genomic markers for *M. leprae* (*16S rRNA* and RLEP) and the human genomic marker (*18S rRNA*) was used for the optimization of qPCR reactions on the portable equipment (Q3-Plus). Paired evaluations were also performed on the standard equipment (Quantstudio 5 –QS-5). The lyophilized synthetic DNA was constituted at 10 fg/μL (equivalent to $10^4$ copies/μL) for *16S rRNA*, RLEP, and *18S rRNA*. To obtain the sample at $10^5$ copies/μL, the gBlock was amplified by qPCR, and its product was purified using the QIAquick PCR Purification Kit (Qiagen, Germany). The concentration was determined by interpolating Cycle threshold (Ct) values on the standard curve. When necessary, samples were diluted in Tris-EDTA (TE) buffer (pH 8.0) for standard curve analyses.

### *M. leprae* cells

*M. leprae* cells (Thai-53, at $10^6$ cells/mL) obtained from nude mice footpads were kindly provided by Dr. Patricia Sammarco Rosa from Lauro de Souza Lima Institute (Bauru, São Paulo, Brazil). The cells were diluted at 1:10 in Tris-EDTA (TE) buffer (pH 8.0) for the construction of the standard curve.

## Clinical samples

The clinical samples were collected based on the occurrence of attendance at the Souza Araújo Clinic from the Oswaldo Cruz Foundation (Rio de Janeiro, Brazil). Skin biopsy collections were performed using a 3 mm surgical punch and stored in 70% ethanol until sample processing. Following clinical assessment, samples were characterized according clinical symptomatology, considered the "gold standard" criterion in this study. Established protocols were employed to assist in diagnosis, including histopathology, bacilloscopy, and singleplex qPCR (*16S rRNA*). Cases were classified according to WHO guidelines as multibacillary (MB), paucibacillary (PB), or other dermatoses (OD) [18].

This study utilized 115 clinical skin biopsy samples, comprising 41 MB, 25 PB, and 49 OD (S1 Table). Of these, 95 (22 PB, 34 MB, and 39 OD) were used in optimization of Q3-Plus. For evaluation of a simplified DNA extraction protocol, 53 (14 MB, 9 PB, and 30 OD) skin biopsy samples were used. The stages of the present study were schematically outlined, with indications of the specific sample groups used in each step (Fig 1).

For the optimization of qPCR reactions on the portable equipment (Q3-Plus), DNA from the 95 samples were exclusively extracted by a commercial kit (Qiagen, Germany).

## Determination of the simplified extraction DNA protocol

The determination of the optimal lysis solution (S2 Table) involved 21 samples of 3 mm swine skin commercially acquired from a local butcher's shop [19,20,21]. Subsequently, the purification and elution steps were also evaluated, as outlined in S1 Fig. Analyses were conducted in two ways: I)Visually, by observing changes in solution turbidity and reduction of skin fragments; II) Using qPCR for the mammalian *18S rRNA* gene.

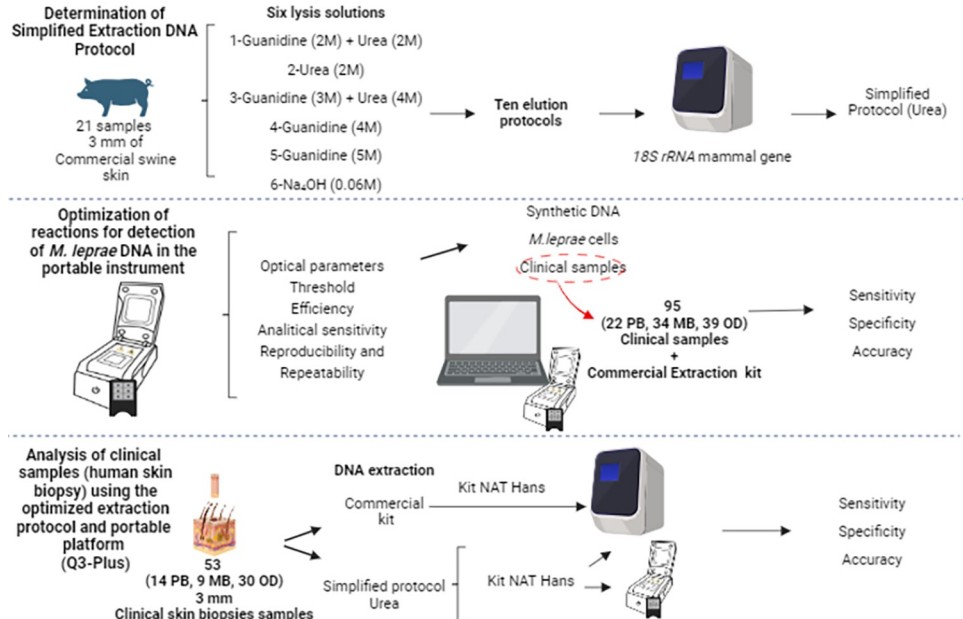

**Fig 1. Schematic representation of the optimization steps of reactions in portable equipment, and the development of the simplified extraction protocol, including the categories of sample involved in each step.** (Credit: Created by the author in BioRender.com).

## Analysis of clinical samples (human skin biopsy) using an optimized extraction protocol and portable platform (Q3-Plus)

Upon establishing the optimal DNA extraction protocol from skin biopsies, a subsequent assessment of 53 clinical human samples (14 MB, 9 PB, and 30 OD) was undertaken. To refine and appraise the streamlined protocol, the 3 mm of biopsy specimen was transferred to a 1.5 mL microtube, containing 146 μL of denaturing solution (8 M urea solution), 14.6 μL of proteinase K (PK) (20 mg/μL) (Roche, Germany), 366 μL of phosphate-buffered saline (PBS) (5.8 mM), and 73.4 μL of nuclease-free water (SOLUTION 2- S2 Table). Next, employing a sterile microtube pestle, the sample was meticulously macerated and then subjected to a 30-minute incubation at 56˚C in a thermal block. Additional rounds of maceration and vortex agitation were performed at ten-minute intervals. Subsequently, 200 μL of the resulting supernatant was meticulously deposited onto an FTA Elute Micro Card, where it was diligently preserved at ambient temperature (21–23˚C) until complete desiccation.

For elution, a 6 mm disc punch of the FTA containing the lysed material underwent two washes with 500 μL of nuclease-free water and then incubated with 100 μL of TE buffer (pH 8.0) at 95˚C for 5 minutes in a thermal block.

## Optimization of reactions for detection of *M. leprae* DNA using the portable instrument

For the optimization of reactions on the Q3-Plus equipment, adjustments to optical parameters, thresholds and cutoffs were necessary. This determination was based on standard curve analyses using synthetic DNA from *M. leprae*, DNA from *M. leprae* cells, DNA from 95 (22 PB, 34 MB, and 39 OD) clinical samples extracted by commercial kit and negative controls. These modifications aimed to enhance the sensitivity of the reactions.

Reactions for detection of *M. leprae* targets (*16S rRNA* and RLEP) and the human internal control (*18S rRNA*) were performed on the Q3-Plus instrument, using the GoTaq Probe qPCR Master Mix (2X) (Promega, USA) and the same oligonucleotides described by our colleagues [12], which are used in the NAT Hans kit (IBMP, Brazil). For the *16S rRNA* target, forward and reverse oligonucleotides concentration were 0.75 μM and 0.3 μM for the probe; for the RLEP target, forward and reverse oligonuleotides concentrations were 0.4 μM, and 0.2 μM for the probe; for the *18S rRNA* target, forward and reverse oligonucleotides were 0.2 μM, and 0.1 μM for the probe. It was necessary to replace Cy5 fluorophore with ROX, which is the probe used for the detection of the *18S rRNA* gene, due to specificities of the instrument. On the portable platform, the reactions were standardized to a final volume of 5 μL, containing 2 μL of DNA.

Optical parameters for the FAM channel were exposure time 1 second, gain 14, and light power 8; for the VIC channel, parameters were exposure time 1 second, gain 14, and light power 9; finally, for the ROX channel, optimized parameters were exposure time 1 second, gain 15, and light power 7. The baseline was defined automatically by the instrument software (Q3-Plus V2 Suite, version 4.0, ST Microelectronics), and the established threshold was 36 arbitrary units of fluorescence (a.u.) for the target *16S rRNA*, 150 a.u. for RLEP and 21 a.u. for *18S rRNA*.

Cycling conditions were 95˚C for 2 minutes, 45 cycles of 95˚C for 15 seconds, and 64˚C for 1 minute. The total reaction time on the portable equipment was approximately 1 hour and 40 minutes.

A Bland-Altman analysis was performed [22], and the average Ct value difference between different equipments for the targets were added to the values of cutoff already established in

the NAT Hans kit. As a result, for the reactions analyzed in Q3-Plus, the Ct cutoffs established for the *16S rRNA* target was 36.9 cycles, and for the RLEP target it was 39.6 cycles.

Samples showing amplification in only one of the targets (*16S rRNA* or RLEP) were categorized as "indeterminate", requiring further testing and follow-up with the patient.

### Commercial DNA extraction

DNA extraction from skin biopsy samples (115 samples) (3 mm) stored in 70% alcohol was performed using DNAeasy Blood and Tissue extraction kit (Qiagen, Germany) according to the manufacturer protocol. DNA concentration was estimated using NanoDrop (Thermo-Fisher Scientific, Waltham, MA, USA) and samples were immediately stored at -20˚C after concentration estimation.

### Experimental conditions for multiplex qPCR (NAT Hans) in standard equipment: QuantStudio 5 (QS-5)

Detection of the two *M. leprae* targets (*16S rRNA* and RLEP) and the human internal control (*18S rRNA*) was performed as previously described by our colleagues [12]. Reactions were performed in a standard benchtop instrument (QuantStudio-5, Applied Biosystems, USA) using the NAT Hans kit and were analyzed following the manufacturer's instructions. Reactions' final volume was 25 μL, containing 5 μL of DNA. Cycling conditions used in QuantStudio-5 were 95˚C for 10 minutes, 45 cycles of 95˚C for 15 seconds, and 60˚C for 1 minute.

For the reactions analyzed on QuantStudio-5, the cutoff Ct values established for the 16S rRNA target were 35.5 cycles, and for the RLEP target they were 34.5 cycles. Samples that showed amplification only in the RLEP target below or equal to Ct 34.5 were classified as "indeterminate".

Positive samples showed amplification for both targets with Ct values equal to or below the cutoff. Negative samples had Ct values above the cutoff or did not show amplification for any of the targets.

### Standard curve and analytical sensitivity on Q3-Plus and Quantstudio-5 equipment

The efficiency and determination of the detection limit for the qPCR reactions on the portable Q3-Plus instrument were obtained through a standard curve with logarithmic scale dilutions using synthetic DNA or purified *M. leprae* DNA (referred to as equivalent-genomes). Efficiency for the reactions was calculated by substituting the slope value of the linear regression line into the efficiency formula ($E = (10^{-1/\text{slope})-1}.*100$), following the MIQE Guidelines [23].

The number of equivalent genome copies was estimated by interpolating Ct values obtained from the analysis of purified *M. leprae* DNA using the equation established after linear regression of the curve performed with synthetic DNA, considering known concentrations and the number of copies (10 fg/μL equivalent to $10^4$ copies/μL of synthetic DNA). For the analysis of the analytical sensitivity of the reaction, seven dilution points were considered, with a higher number of replicates (nine or ten) for the lower concentrations [24].

As a reference, identical analyses were performed using the standard QS-5 equipment.

### Statistical analysis

Diagnostic parameters such as sensitivity, specificity, and accuracy were obtained using clinical evaluation as the gold standard. Analyses of reaction efficiency, reproducibility, and repeatability were conducted for the Q3-Plus equipment. The detection limit with 95% confidence interval ($LOD_{95\%}$) was calculated using the Probit model using RStudio version 4.1.0 software

**Table 1. Ct results and avarage Cts from evaluation of lysis solutions for simplified DNA extraction protocol, based on mammalian *18S rRNA* target detection by qPCR in swine skin using QS-5 equipment.** Protocol numbers are summarized as 1—Guanidine (2 M) and Urea (2 M); 2 –Urea (2 M); 3 –Guanidine (3 M) and Urea (4 M); 4 –Guanidine (4 M); 5 –Guanidine (5 M); 6 –NH₄OH (0.06 M); NC–Negative control.

| Evaluated protocols–Molecular detection of the *18S rRNA* target | | | | | | | |
|---|---|---|---|---|---|---|---|
| **Protocols** | **1** | **2** | **3** | **4** | **5** | **6** | **NC** |
| **Ct** | 30.75 | 21.87 | 23.69 | 24.77 | 28.11 | 23.93 | 29.50 |
| | 26.57 | 21.16 | 23.11 | 23.03 | 28.81 | 31.68 | 29.78 |
| | 29.08 | 22.08 | 24.26 | 23.62 | 27.76 | 25.01 | 29.27 |
| **Mean Ct** | **28.80** | **21.70** | **23.69** | **23.81** | **28.23** | **26.87** | **29.52** |

(downloaded from http://www.Rproject.org/). Evaluation of the agreement between the different instruments was carried out using the Bland-Altman analyses [22]. ANOVA was performed to assess statistical differences in analysis of reproducibility and repeatability.

This study follows the STARD guidelines for reporting diagnostic accuracy studies [25]. The minimum information for publication of quantitative real-time PCR experiments (MiQE) [23] are presented as Supplemental Material (S3 and S4 Tables).

## Results

### Determination of the simplified extraction DNA protocol

**Evaluation of lysis solutions for swine skin models.** The best performing lysis solution was determined on swine skin using six different solutions. Analyses were conducted in two ways: I) visually, and II) by detection of the mammalian *18S rRNA* target on the standard equipment QS-5 [19,20,21]. Table 1 summarizes all six protocols that were evaluated and the corresponding mean Ct for qPCR detection of the mammalian *18S rRNA* gene in QS-5. A mixture consisting of urea (2 M), proteinase K (0.5 mg/mL), and PBS pH 7.4 (3.5 mM) yielded the best results in the visual evaluations regarding the turbidity of the solution and reduction/dissolution of the fragment of skin, and in the detection of the *18S rRNA* gene by qPCR in terms of fluorescence amplitude and Ct. In the porcine tissue used as a comparative model, the mean Ct was 18.74 (range 17.45 to 19.40 cycles) in the detection of the *18S rRNA* gene.

### Analysis of clinical samples (human skin biopsy) using the developed extraction protocol and portable platform (Q3-Plus)

The entire newly developed method, from DNA extraction using the simplified protocol with urea solution (SOLUTION 2 –S2 Table) (Fig 2) to qPCR analysis for the detection of *M. leprae* targets on the portable platform, was evaluated using 53 clinical skin biopsy samples from patients with leprosy (MB and PB) and patients with other dermatoses (OD).

During visual assessment while extracting genetic material, a reduction in the skin biopsy fragment or its complete dissolution was observed, resulting in a visibly altered solution turbidity (S2 Fig). Successful detection of the *18S rRNA* gene was also achieved. In reactions conducted on the standard equipment (QS-5), the mean Ct was 21.12 cycles (ranging from 17.24 to 28.43 cycles), while on the Q3-Plus equipment, the mean Ct was 22.46 cycles (ranging from 19.02 to 30.54 cycles) (S5 Table).

In a Bland-Altman test conducted based on the same group of samples, with 95% confidence the *16S rRNA* target, the mean threshold cycle variation between paired samples was 1.29 cycles higher for the Q3-Plus equipment (S3 Fig). For the detection of the *M. leprae* RLEP target, a mean variation of 4.44 cycles above was observed for the Q3-Plus equipment (S4 Fig). Finally, for the *18S rRNA*, a mean variation of 1.34 threshold cycles was observed (S5 Fig).

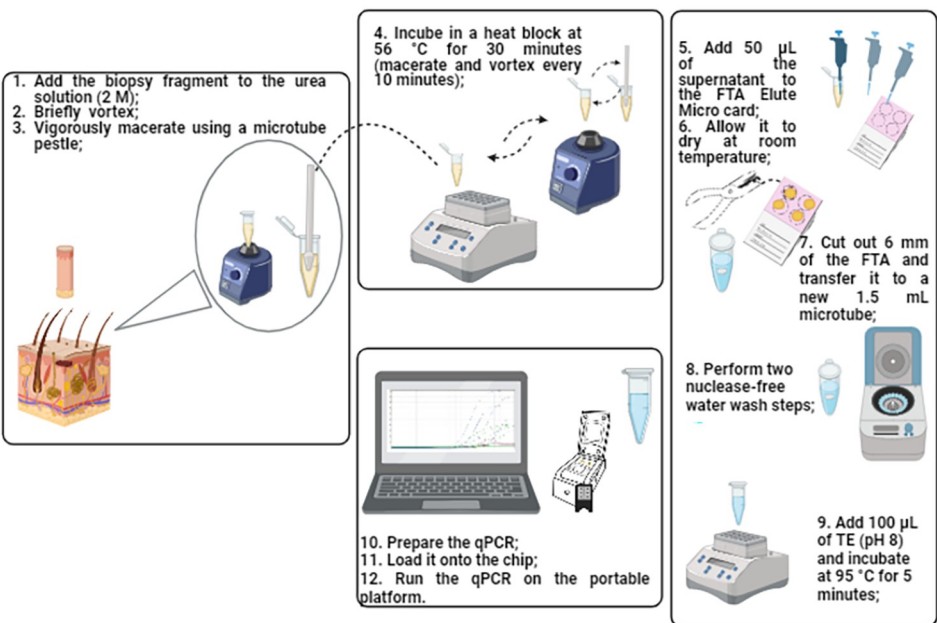

**Fig 2. Schematic representation of the simplified DNA extraction protocol for *Mycobacterium leprae* from clinical skin biopsy samples.** (Credit: Created by the author in BioRender.com).

qPCR experiments using the NAT Hans kit on the Q3-Plus equipment yielded positivity for *M. leprae* in 26% (14/53) of the samples, while 55% (29/53) of the samples tested negative, and 19% (10/53) of samples yielded indeterminate results. Among the 22 samples with discordant results between the clinical evaluation (gold standard method) and the developed method (simplified extraction + analysis using a portable device), 8 samples previously classified as PB and 4 previously classified as MB yielded negative or indeterminate results with the simplified method. Additionally, of the 10 samples classified as OD, 3 had positive results and 7 were indeterminate with the simplified method.

For the analyses conducted on the standard equipment (QS-5), 30% (16/53) tested positive for the *M. leprae* agent, while 68% (36/53) yielded negative results. On this equipment, 2% (1/53) of the samples produced indeterminate results. In these analyses, 13 samples yielded discordant results when compared with clinical outcome: 7 samples previously classified as PB and 3 classified as MB yielded negative or indeterminate results, and 3 samples previously classified as OD also presented positive results when processed with the simplified method.

In the analyses conducted from samples extracted using a commercial kit (Qiagen) and evaluated on standard benchtop equipment (QS-5), 38% (20/53) were positive, 30% (16/53) negative, and 32% (17/53) indeterminate. When compared with clinical diagnostic, 28 samples presented discordant results. It was observed that 7 samples classified as PB and 2 classified as MB yielded negative of indeterminate results. Additionally, of the 19 samples previously classified as OD, 13 presented indeterminate result and 6 presented positive results.

Parameters of sensitivity, specificity, and accuracy of the different protocols according to clinical form are presented in Table 2.

## Optimization of reactions for detection of *M. leprae* DNA in the portable instrument

**Optical parameters and efficiency of reactions.** Different values of optical parameters were evaluated in the reactios optimization in the Q3-Plus instrument. For FAM channel,

**Table 2. Comparison of molecular diagnostic parameters in 53 clinical biopsies in evaluated simplified extraction protocol and commercial protocol kit across different platforms (Q3-Plus and Quantstudio 5).** PB: paucibacillary; MB: multibacillary.

| Parameters | Simplified Protocol NAT Hans Q3-Plus | Simplified Protocol NAT Hans Quantstudio-5 | Commercial extraction NAT Hans Quantstudio-5 | Simplified Protocol NAT Hans Q3-Plus | | Simplified Protocol NAT Hans Quantstudio-5 | | Commercial extraction NAT Hans Quantstudio-5 | |
|---|---|---|---|---|---|---|---|---|---|
| | PB + MB | PB + MB | PB + MB | PB | MB | PB | MB | PB | MB |
| Sensitivity | 55% | 59% | 74% | 13% | 83% | 22% | 85% | 33% | 92% |
| Specificity | 87% | 90% | 65% | 87% | 87% | 90% | 90% | 65% | 65% |
| Accuracy | 72% | 77% | 69% | 68% | 86% | 74% | 88% | 57% | 77% |

parameters were set as exposure time 1, gain 14, and light power 8. For VIC channel, parameters were exposure time 1, gain 14, and light power 9. For the channel ROX channel, optimized parameters were exposure time 1, gain 15, and light intensity 7.

Fluorescence threshold for each target was set at 36 a.u. for the *16S rRNA* target (FAM), 150 a.u. for the RLEP target (VIC) of *M. leprae*, and 21 a.u. for the *18S rRNA* target (ROX).

The efficiency based on a dilution curve ($10^5$–$10^0$ copies/µL) of synthetic DNA (gblock) was 109% for the *16S rRNA* and 108% for RLEP on the Q3-Plus. The standard QS-5 equipment showed 102% efficiency for *16S rRNA* and 94% for RLEP (S6 Fig)(S6 Table).

Reactions containing $10^6$ to $10^0$ genome-equivalent/µL of *M. leprae* cells yielded efficiencies of 131% for the *16S rRNA* target and 105% for the RLEP target were observed in the Q3-Plus equipment. The QS-5 equipment showed efficiencies of 101% for *16S rRNA* and 100% for RLEP (S7 Fig)(S6 Table).

**Analytical sensitivity.** To assess the analytical sensitivity of the newly developed method in the Q3-Plus and QS-5 instruments, the limit of detection with 95% confidence interval ($LOD_{95\%}$) was determined. Using synthetic DNA, the $LOD_{95\%}$ was 13.86 copies/µL for both the *16S rRNA* and RLEP targets on the portable instrument. On the QS-5 instrument, the corresponding values were 12.45 copies/µL for *16S rRNA* and 20.44 copies/µL for RLEP. When purified *M. leprae* DNA was used, the $LOD_{95\%}$ on the portable device was 113.31 genome-equivalents/µL for the *16S rRNA* gene and 17.70 genome-equivalents/µL for the RLEP (Fig 3). On the standard equipment (QS-5), the values were 205.26 genome-equivalents/µL for *16S rRNA*, and 15.34 genome-equivalents/µL for the RLEP.

**Reproducibility and repeatability in the portable equipment.** Intra- and inter-operator reproducibility tests for the Q3-Plus equipment showed coefficients of variation below 10% for the *16S rRNA* and RLEP targets, indicating non-significant differences (S7 Table). In the inter-operator analyses, the coefficient of variation values between concentrations was between 0.05

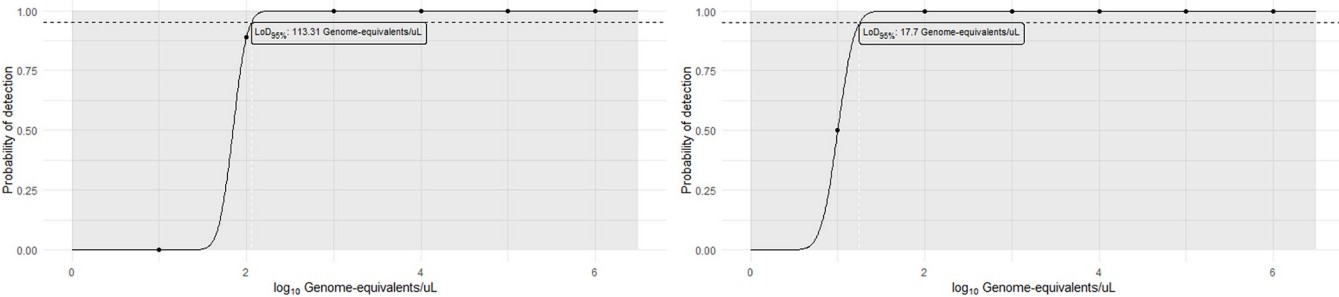

**Fig 3. Analytical sensitivity of qPCR reactions for the targets *16S rRNA* and RLEP of *Mycobacterium leprae* performed on the portable Q3-Plus equipment.** (A) LOD95% *16S rRNA*; (B) LOD95% RLEP. (Created by the author.)

**Table 3. Comparison of molecular diagnostic parameters in 95 clinical biopsies evaluated in commercial extraction protocol kit across different platforms (ABI7500, QuantStudio-5 and Q3-Plus platforms).** PB: paucibacillary; MB: multibacillary.

| Parameters | Commercial extraction | | | | | | | | |
|---|---|---|---|---|---|---|---|---|---|
| | NAT Hans Q3-Plus | NAT Hans Quantstudio 5 | Singleplex *16S rRNA* ABI7500 | NAT Hans Q3-Plus | | NAT Hans Quantstudio 5 | | Singleplex *16S rRNA* ABI7500 | |
| | PB + MB | PB + MB | PB + MB | PB | MB | PB | MB | PB | MB |
| Sensitivity | 73% | 76% | 73% | 19% | 100% | 25% | 100% | 22% | 100% |
| Specificity | 85% | 76% | 84% | 85% | 85% | 76% | 76% | 84% | 84% |
| Accuracy | 78% | 76% | 77% | 63% | 92% | 60% | 88% | 64% | 91% |

and 3.09% for the *16S rRNA* gene and 0.55 to 3.10% for the RLEP. The analysis of variance (ANOVA) performed for the results obtained by the different operators, assuming a confidence interval of 95%, corroborated the previous analyses, showing no significant inter-operator difference (p-value = 0.994 for *16S rRNA*, and p-value = 0.992 for RLEP) (S7 and S8 Tables).

**Evaluation of the portable qPCR using DNA from skin biopsies.** To determine the cutoff to reactions on Q3-Plus, the mean difference observed between the instruments for the different targets was added to the NAT Hans kit values. Therefore, for the reactions analyzed in Q3-Plus the Ct cutoffs established for the *16S rRNA* target was 36.9 cycles, and for the RLEP target it was 39.6 cycles. Samples with Ct values below these cutoffs were classified as positive; those above or with no amplification were negative. Samples that exhibited amplification in only one of the targets (*16S rRNA* or RLEP) were classified as indeterminate, necessitating retesting and patient follow-up. Of the 95 samples, 43% (41/95) were positive, 43% (41/95) negative, and 14% (13/95) indeterminate for *M. leprae* DNA. The Ct scores for each of the analyzed samples are presented in S9 TABLE.

For the standard equipment (QS-5), which served as the benchmark, the NAT Hans kit values were used according to the recommended cutoff in the commercial kit protocol. Results showed 48% (46/95) positive, 40% (38/95) negative, and 12% (11/95) of the indeterminate for *M. leprae* DNA.

A difference in the Ct value was observed for the same 95 samples extracted by commercial kit when analyzed on different instruments. The mean Ct difference for the same samples between the instruments resulted in an increase of 1.40 cycles for the Q3-Plus instrument to *16S rRNA* target and an increase of 5.15 cycles for the RLEP target when compared to that obtained in the standard instrument analysis (S8 and S9 Figs). In the internal control (*18S rRNA*) there was a mean increase of 1.42 cycles on Q3-Plus analysis (S10 Fig). Cycle values above these means were mainly observed in samples classified as PB, where a lower concentration of the target DNA is expected.

Parameters of sensitivity, specificity, and accuracy of the different equipments according to clinical form are presented in Table 3.

## Discussion

The utilization of screening tests in settings with restricted resources, along with their contribution to active case detection, holds paramount significance in achieving reduced leprosy incidence rates [4,26]. Considering the absence of a definitive gold standard test, the inherent constraints of adjunct assays, and the clinical nuances of the disease, the advancement of assays characterized by high sensitivity and specificity on portable platforms, accompanied by cost reduction and technique streamlining, facilitates the adoption of preventive and control

interventions within the disease transmission continuum. Thus, our study is the laboratory validation of a prototype, contributing towards the development of portable tools that might improve the accessibility of the leprosy diagnosis to vulnerable populations.

Among the evaluated extraction protocols, certain chaotropic agents such as urea, guanidine, and ammonium hydroxide ($NH_4OH$) were preferred, and the usage of FTA cards embedded in detergents aided in the isolation and purification steps of the genetic material. Due to their affinity for the cellulose fibers in the card, DNA recovery was feasible after simple washing steps [27]. The detergents enable cellular lysis and protein denaturation, further facilitating DNA exposure and recovery. These filter paper cards were developed to streamline sample transport and storage at room temperature, while ensuring DNA viability for molecular analyses of interest [28,29] as well as requiring minimal space for storage and having low risk of cross-contamination [30].

Among the evaluated chaotropic reagents, urea yielded the best results in dissolving the skin fragments, which may be linked to improved activity of proteinase K facilitated by high concentration of urea ass well as to improves preservation of DNA in a medium full of nucleases, as reported in previous studies [31,32]. Improved tissued dissolution was also reflected in the qPCR detection of the human *18S rRNA* target, as the Ct values were lower than those yielded by other protocols and more consistent across replicates.

The search for *M. leprae* targets (*16S rRNA* and RLEP) in clinical human skin biopsies samples showed a loss of sensitivity when using the simplified protocol-extracted samples, as evidenced by the increased Ct value when compared to the standard test performed with sampled extracted by commercial method. This decrease in sensitivity affects the detection of paucibacillary cases. Other studies using qPCR showed variation between 36.4% and 85% in sensitivity for PB cases using different targets and biological materials. In these studies, lower sensitivity is also observed for PB cases, showcasing the intrinsic limitation in detecting this clinical form [11,14,33,34,35].

Interestingly, higher specificity was observed in the detection of *M. leprae* in clinical samples processed using the simplified method and analyzed on the portable platform. Higher specificity is desirable in a screening test as it demonstrates the test's ability to correctly identify individuals who do not have leprosy. By reducing the detection of false positives, it ensures more accurate case screening and better allocation of resources for the diagnosis and treatment of those who are truly positive, especially in low resource areas [13].

Despite our best efforts, some discordant results between clinical diagnosis (gold standard) and analyses performed on different platforms were found. Even though some samples were classified as PB or MB, having a bacilloscopy index (iB) of less than 2 affected molecular detection, as expected. The low bacterial load may have impeded the detection of genetic material [12]. Additionally, the method employed uses an extraction process that does not rely on silica columns, which may result in residual reagents affecting the accuracy of detection.

Regarding the samples classified as OD according to the clinical evaluation, but were positive or indeterminate in the molecular tests, some factors need to be considered; The final outcome of the patient may evolve over a year of follow-up, as the initial phase of the disease can be asymptomatic or present with subtle clinical signs. Although clinical evaluation is crucial for diagnosis, positive and indeterminate results in molecular tests justify the need for ongoing monitoring of these cases for at least 12 months. Molecular test positivity may occur even in patients initially categorized as OD due to unclear clinical symptoms, but who will become PB or MB later on, after the onset of the symptoms, as well as in patient short of following treatment, or in immunocompetent patients who were positive but have cleared any live bacillus [35].

The optimization of qPCR reactions on a portable analysis platform (Q3-Plus) was established using the oligonucleotides included in the first Brazilian diagnostic kit for leprosy,

approved by ANVISA (kit NAT Hans–IBMP). The molecular kit involves the detection of two targets (*16S rRNA* e RLEP) from *M. leprae* and an internal control target (*18S rRNA* from mammals) in the reaction. Detection of the *18S rRNA* target ensures the presence of human DNA, validating the success of the collection, extraction, and qPCR steps. Instead, the *16S rRNA* target, being a single-copy, conserved gene, enhances the test's sensitivity and specificity when combined with RLEP in the reaction [12]. The Q3-Plus equipment, being a point-of-care device due to its compact dimensions (7x14x8.5 cm), uses reaction volumes of 5 µL, requiring minor adjustments to oligonucleotide concentrations and master mix composition. Nevertheless, the reactions exhibited excellent efficiency values, closely approaching those of the standard equipment Quantstudio-5, highlighting the applicability of the platform in analytical contexts. Furthermore, it is an equipment that has a user friendly interface [36] and has been used for the detection of several other pathogens, such as *T. cruzi* and *Plasmodium* spp. [37], *M. tuberculosis* [38], and *C. thachomatis* [39].

The efficiency of reaction is linked to the exponential amplification of the target material throughout the analysis. Factors such as the purity and concentration of the target in the sample and reagents, as well as the final volume of the qPCR reaction, are known determinants of the technique's efficiency [40]. Although the final reaction volume in the Q3-Plus equipment is five times smaller than that used in reactions analyzed in the standard equipment, the efficiency values remained close to 100%, which is desirable for this type of analysis [40]. Previous studies on reaction optimization for molecular detection of *Plasmodium* spp., *Trypanosoma cruzi*, and *Mycobacterium tuberculosis* had already noted higher efficiencies in the portable device when compared to the standard (ABI7500) [38,37], as observed in the current study. This reinforces the possibility of this parameter being an anticipated trait of the Q3-Plus equipment. However, it is noteworthy that, although efficiencies were higher than desired, the limits of detection of all reactions remained within clinically-relevant values.

The determination of the fluorescence threshold is crucial to prevent non-specific amplifications or other inferences from causing false results in molecular qPCR tests. This parameter can be established either through numerical analysis or visually by the operator [41]. Despite its subjective nature, in this study, the threshold was manually set by observing the fluorescence amplitude in known samples. Therefore, the higher Ct values observed for the RLEP marker are justified, as the threshold defined for this target was 150 arbitrary units (a.u.).

The reactions from synthetic DNA (gBlock) in both instruments exhibited linearity up to $10^1$ copies/µL. Beyond this range, amplifications started to occur stochastically down to $10^{-1}$ copies/µL. In the analyses of equivalent genome/µL of *M. leprae*, linearity in the instruments extended up to $10^2$ equivalent genomes/µL; however, beyond this range, the Q3-Plus instrument lost analytical sensitivity for *16S rRNA* target, while the QS-5 began to exhibit random amplifications down to $10^{-1}$ equivalent genomes/µL. For the RLEP target, amplifications persisted beyond the linear range in both instruments. Given that this is a multicopy target, higher analytical sensitivity is expected [42]. The loss of reaction linearity implies the occurrence of random amplifications, which could interfere particularly in cases classified as paucibacillary due to low bacillary load [43].

Through the analysis of the limit of detection (LOD$_{95\%}$), it was possible to confirm greater sensitivity for the RLEP target. Reactions with purified *M. leprae* DNA yielded LOD values of 113.31 genome-equivalents/µL for the *16S rRNA* gene and 17.70 genome-equivalents/µL for RLEP on the Q3-Plus instrument. In a previous study by our colleagues [12], the authors reported LOD values of 126 genome-equivalents/reaction for the *16S rRNA* and 1.3 genome-equivalents/reaction for the RLEP target, using the NAT Hans kit in a standard thermocycler (ABI7500). This proximity of values ensures the good analytical sensitivity of the portable instrument. It is important to note that the observed analytical sensitivity of 13.86 copies/µL

for both targets, *16S rRNA* and RLEP, was the same when using synthetic DNA, which contains a single copy of each target. However, this does not reflect the situation in biological samples containing the *M. leprae*, where higher analytical sensitivity for RLEP is expected due to its multicopy nature (approximately 36 copies per genome) [12,42].

Analysis intra and inter-operator show very good results. All coefficients of variation were found to be below 5%. The three data points from intra-operator assessments that exceed this variation correspond to the lowest concentrations of the target DNA, falling outside or at the limit of the reaction, where the probability of amplification decreases and becomes stochastic.

In Bland-Altman analyses on pre-characterized clinical samples, a mean variation of approximately 1.40 cycles higher for the *16S rRNA* target was observed on the Q3-Plus equipment compared to QS-5. For the RLEP target, the observed variation was approximately 5.15 cycles higher in the Q3-Plus evaluated samples. These values closely align with those reported by our colleagues [37] from the optimization of the Q3 equipment for molecular detection of *T. cruzi* and *Plasmodium* spp. The disparity noted by these authors amounted to an increase of 2 to 4 cycles in Q3-Plus reactions.

The results obtained from optimization of reactions on the portable platform, concerning optical parameters, reaction efficiency, and analytical sensitivity, were confirmed in pre-characterized clinical samples extracted using the commercial kit (Qiagen). Analyses of sensitivity, specificity, and accuracy comparable to those of established qPCR tests on the standard equipment confirm the applicability of point-of-care testing. The occurrence of false negatives is observed especially in paucibacillary cases [44,45]. The low bacterial load hinders the detection of these cases. However, qPCR is still considered the best technique to be used as a screening test due to its high sensitivity and specificity, particularly in detecting PB cases [44,46].

The results of the present study demonstrate the need for further protocol optimization to improve the detection of PB cases. However, the results are promising. The complete technological platform might be used as an auxiliary tool in detecting leprosy cases in remote regions and vulnerable populations. Social vulnerability, particularly observed in areas with low infrastructure, is relevant in perpetuating the disease transmission chain [47,48].

Furthermore, the possibility of using molecular tests may reduce recurring misdiagnoses in leprosy [2,14]. Ensuring diagnosis for all populations is essential, and decentralizing access to it, as facilitated by active case finding, is pivotal for leprosy to discontinue being considered a public health problem in Brazil [49]. The Global Leprosy Strategy 2021–2030, published by the World Health Organization (WHO) [50], aims to eliminate the disease by interrupting transmission. However, there is a consensus that this goal will only be achieved with the improvement of the current strategies of complementary diagnosis, with an emphasis not only on developing strategies to increase the sensitivity of current tests, but also to increase access to available tests. The portable platform serves as a tool that, by adhering to the principles of the point-of-care testing concept, can contribute to overcoming some of the limitations in leprosy diagnosis.

As a limitation of this study, it should be considered that the samples were collected based on their occurrence in the clinic. The final clinical outcome of cases will be determined after one year of follow-up. Additionally, molecular test positivity can also occur in cases under treatment, where residual bacillus DNA may be present.

## Conclusion

The newly developed simplified sample processing protocol yielded qPCR-ready DNA, as shown by detection of the human *18S rRNA* gene. Aiming the development of a truly portable molecular testing platform, reactions for detection of *M. leprae* DNA were optimized on a

portable device. The complete technological solution (DNA extraction using a simplified protocol and qPCR analysis on the portable instrument) yielded excellent results for screening multibacillary and negative cases. The results presented here constitute the laboratory validation of a prototype designed for aiding leprosy diagnosis in field conditions. The prototype shows promising potential as an auxiliary screening tool for healthcare professionals in low resource or remote areas.

## Supporting information

**S1 Table. Information regarding clinical samples, gender, age and clinical diagnosis collected at the Hansen's disease Laboratory of Oswaldo Cruz Institute–Fiocruz–RJ.** Key: PB–Paucibacillary; MB–Multibacillary; OD–Other dermatoses.
(XLSX)

**S2 Table. Lysis solutions were assessed in protocols aimed at developing a simplified extraction method.** Shown are the protocols and reagents with their respective concentrations and volumes necessary for the preparation of lysis solutions evaluated in DNA extraction from 3 mm skin fragment of swine skin; *PBS: Phosphate Buffered Saline; NH4OH: ammonium hydroxide; Tx100: Triton X-100; SDS: Sodium Dodecyl Sulfate. Initially, all solutions were tested using porcine skin as the experimental model. Following the identification of the most effective lysis solution, which was solution 2, we proceeded to evaluate using skin biopsies from human patients.
(XLSX)

**S3 Table. STARD checklist.**
(DOCX)

**S4 Table. MIQE checklist.**
(DOCX)

**S5 Table. qPCR analysis of skin biopsies from suspected leprosy patients extracted using the simplified protocol.** Ct scores are observed for each target (*16S rRNA*, RLEP, and *18S rRNA*) obtained in qPCR analyses on the Q3-Plus and Quantstudio-5 equipment. Clinical outcome data (clinical evaluation) are also presented.
(XLSX)

**S6 Table. Ct values obtained from qPCR analyses using synthetic DNA and DNA from *Mycobacterium leprae* cells for standard curve determination on Q3-Plus and Quantstudio 5 instruments.**
(XLSX)

**S7 Table. Repeatability and Reproducibility analyses, including the respective coefficients of variation, were determined in qPCR reactions in synthetic DNA to detect targeting *Mycobacterium leprae 16S rRNA* and RLEP on the Q3-Plus equipment.**
(XLSX)

**S8 Table. Ct Values from Repeatability and Reproducibility tests used for analyses.**
(XLSX)

**S9 Table. qPCR results of skin biopsies from suspected leprosy patients extracted using commercial kit protocol in optimization analyses.** Ct scores are observed for each target (*16S rRNA*, RLEP, and *18S rRNA*) obtained in qPCR analyses on the Q3-Plus and Quantstudio-5 equipment. Clinical outcome data (clinical evaluation) are also presented.
(XLSX)

**S1 Fig. Elution protocols from FTA cards evaluated in the extraction of *Mycobacterium leprae* DNA from skin biopsy samples.** (Created by the author)
(TIF)

**S2 Fig. Demonstration of the simplified extraction protocol in clinical biopsy sample.** After each 10-minute step, a reduction in skin fragments and a change in the turbidity of the solution were observed. (Created by the author)
(TIF)

**S3 Fig. Bland-Altman *16S rRNA* (Evaluated simplified extraction protocol in clinical samples analyzed in Q3-Plus).** Bland-Altman analysis in *16S rRNA* target. The mean difference was 1.29 cycle of threshold between the equipment in 95% confidence interval. The upper limit of agreement was 12.39 and the lower limit of agreement was -9.81. (Created by the author)
(TIF)

**S4 Fig. Bland-Altman RLEP (Evaluated simplified extraction protocol in clinical samples analyzed in Q3-Plus).** Bland-Altman analysis in RLEP target. The mean difference was 4.44 cycle of threshold between the equipment in 95% confidence interval. The upper limit of agreement was 11.91 and the lower limit of agreement was -3.04. (Created by the author)
(TIF)

**S5 Fig. Bland-Altman *18S rRNA* (Evaluated simplified extraction protocol in clinical samples analyzed in Q3-Plus).** Bland-Altman analysis in *18S rRNA* target. The mean difference was 1.34 cycle of threshold between the equipment in 95% confidence interval. The upper limit of agreement was 6.90 and the lower limit of agreement was -4.22. (Created by the author)
(TIF)

**S6 Fig. Standard curve of qPCR reactions analyzed by QuantStudio and Q3-Plus equipment for the detection of *Mycobacterium leprae 16S rRNA* and RLEP targets from synthetic DNA.** (A) Reactions on QuantStudio 5; (B) Reactions on portable platform Q3-Plus. Linear regressions were obtained from no less than 4 independent experiments. (Created by the author)
(TIF)

**S7 Fig. Standard curve of qPCR reactions analyzed by QuantStudio and Q3-Plus equipment for the detection of *Mycobacterium leprae 16S rRNA* and RLEP targets using purified DNA from *M. leprae*.** (A) Reactions on QuantStudio 5; (B) Reations on portable platform Q3-Plus. Linear regressions were obtained from no less than 4 independent experiments. (Created by the author)
(TIF)

**S8 Fig. Bland-Altman *16S rRNA* target (Evaluated commercial extraction protocol in clinical samples analyzed in optimization of Q3-Plus).** The mean difference it was 1.40 cycle of threshold between instruments. The upper limit of agreement with 95% confidence interval was 9.40 and the lower limit of agreement was -6.61. (Created by the author)
(TIF)

**S9 Fig. Bland-Altman RLEP target (Evaluated commercial extraction protocol in clinical samples analyzed in optimization of Q3-Plus).** The mean difference it was 5.15 cycle of threshold between the equipment in 95% confidence interval. The upper limit of agreement

was 10.59 and the lower limit of agreement was -0.28. (Created by the author)
(TIF)

**S10 Fig. Bland-Altman *18S rRNA* target (Evaluated commercial extraction protocol in clinical samples analyzed in optimization of Q3-Plus).** The mean difference it was 1.42 cycle of threshold between the equipment in 95% confidence interval. The upper limit of agreement was 3.25 and the lower limit of agreement was -0.40. (Created by the author)
(TIF)

## Acknowledgments

The authors are grateful to the entire team of dermatologists, nurses and technicians that collaborate at the Souza Araújo Clinic from the Leprosy Laboratory at the Oswaldo Cruz Institute. We especially thank Raquel Barbieri, Alexsandro Cruz Barreto and Cristiane Domingues for all the technical and administrative assistance.

## Author Contributions

**Conceptualization:** Milton Ozório Moraes, Fernanda Saloum de Neves Manta, Alexandre Dias Tavares Costa.

**Data curation:** Amanda Bertão-Santos, Fernanda Saloum de Neves Manta.

**Formal analysis:** Amanda Bertão-Santos, Marcelo Ribeiro-Alves, Fernanda Saloum de Neves Manta, Alexandre Dias Tavares Costa.

**Funding acquisition:** Roberta Olmo Pinheiro, Milton Ozório Moraes, Alexandre Dias Tavares Costa.

**Investigation:** Amanda Bertão-Santos, Larisse da Silva Dias, Fernanda Saloum de Neves Manta.

**Methodology:** Amanda Bertão-Santos, Larisse da Silva Dias, Fernanda Saloum de Neves Manta, Alexandre Dias Tavares Costa.

**Project administration:** Roberta Olmo Pinheiro, Milton Ozório Moraes, Alexandre Dias Tavares Costa.

**Resources:** Roberta Olmo Pinheiro, Milton Ozório Moraes, Alexandre Dias Tavares Costa.

**Supervision:** Fernanda Saloum de Neves Manta, Alexandre Dias Tavares Costa.

**Writing – original draft:** Amanda Bertão-Santos, Fernanda Saloum de Neves Manta.

**Writing – review & editing:** Marcelo Ribeiro-Alves, Roberta Olmo Pinheiro, Fernanda Saloum de Neves Manta, Alexandre Dias Tavares Costa.

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
