## [Decision Letter · Decision Letter 0]

22 May 2024

Dear Dr. Costa,

Thank you very much for submitting your manuscript "Validation of the performance of a point of care molecular test for leprosy: from a simplified DNA extraction protocol to a portable qPCR" for consideration at PLOS Neglected Tropical Diseases. As with all papers reviewed by the journal, your manuscript was reviewed by members of the editorial board and by several independent reviewers. In light of the reviews (below this email), we would like to invite the resubmission of a significantly-revised version that takes into account the reviewers' comments. 

We are returning your manuscript with three independent expert reviews. All of the Reviewers acknowledged that a rapid point of care molecular test for the detection of M. leprae in skin samples would be very valuable, but also agreed that the description and presentation of data in the current manuscript need to be improved. 

As suggested by Reviewer 2, the following modifications will improve data presentation and interpretation:

- To evaluate M. leprae DNA extraction methods, authors should extract from a known quantity of M. leprae cells and show a comparison of qPCR data for the different conditions (i.e. lysis buffers) for each of the M. leprae targets.

- To evaluate the accuracy of the new test, authors should report Ct scores for each separate group (MB, PB, and OD) for both of the M. leprae targets. 

In addition, as outlined by Reviewers 1 and 3, the methods and results sections should be shortened and streamlined to highlight the final protocol and diagnostic test evaluation with clinical samples together with a better structured description of the optimization procedures. Please modify the manuscript according to these and several additional suggestions (see below) by the Reviewers before re-submission.

We cannot make any decision about publication until we have seen the revised manuscript and your response to the reviewers' comments. Your revised manuscript is also likely to be sent to reviewers for further evaluation.

Sincerely,

Katharina Röltgen

Academic Editor

Mathieu Picardeau

Section Editor

We are returning your manuscript with three independent expert reviews. All of the Reviewers acknowledged that a rapid point of care molecular test for the detection of M. leprae in skin samples would be very valuable, but also agreed that the description and presentation of data in the current manuscript need to be improved. 

As suggested by Reviewer 2, the following modifications will improve data presentation and interpretation:

- To evaluate M. leprae DNA extraction methods, authors should extract from a known quantity of M. leprae cells and show a comparison of qPCR data for the different conditions (i.e. lysis buffers) for each of the M. leprae targets.

- To evaluate the accuracy of the new test, authors should report Ct scores for each separate group (MB, PB, and OD) for both of the M. leprae targets. 

In addition, as outlined by Reviewers 1 and 3, the methods and results sections should be shortened and streamlined to highlight the final protocol and diagnostic test evaluation with clinical samples together with a better structured description of the optimization procedures. Please modify the manuscript according to these and several additional suggestions (see below) by the Reviewers before re-submission.

Reviewer's Responses to Questions

**Key Review Criteria Required for Acceptance?**

**Methods**

-Are the objectives of the study clearly articulated with a clear testable hypothesis stated?

-Is the study design appropriate to address the stated objectives?

-Is the population clearly described and appropriate for the hypothesis being tested?

-Is the sample size sufficient to ensure adequate power to address the hypothesis being tested?

-Were correct statistical analysis used to support conclusions?

-Are there concerns about ethical or regulatory requirements being met?

Reviewer #1: -Are the objectives of the study clearly articulated with a clear testable hypothesis stated?yes

-Is the study design appropriate to address the stated objectives?yes MIQE guidelines are followed

-Is the population clearly described and appropriate for the hypothesis being tested?yes

-Is the sample size sufficient to ensure adequate power to address the hypothesis being tested? sample size is not calculated

-Were correct statistical analysis used to support conclusions?yes

-Are there concerns about ethical or regulatory requirements being met?yes, although not clear where the pork skin comes from and whether ethical approval is necessary in Brazil for this

Reviewer #2: The objectives of the study are articulated well although the organization and flow could be improved.

The study design involves MB and PB leprosy patients and other dermatoses (OD) but the numbers in these groups changes with different extraction protocols so it is a little hard to follow.

No ethical concerns.

Reviewer #3: the objective of a point of care test to diagnose leprosy using molecular tools is clear and valuable. The authors compared different reagents, buffers for all the steps needs for these diagnostic tools, i.e. extraction from the skin sample, DNA purification, PCR and interpretation. They interpreted their results for measuring the performances to the test done in a lab using a Quantstudio rela time PCR and a kit used widely in Brazil, the NAT Hans kit. 

the sample size of 115 clinical samples was adequate to this study,.

The major comments are the following: the methods are far too detailed for a publication and all the intermediary steps to reach the final protocol need to be deleted or put in an appendix or supplementary materials (e.g. tables and 1 and 2). only the method finally approved and that was used to compare the results, need to be described herein.

**Results**

-Does the analysis presented match the analysis plan?

-Are the results clearly and completely presented?

-Are the figures (Tables, Images) of sufficient quality for clarity?

Reviewer #1: figures lack abbreviation explanations

Are the results clearly and completely presented? can be more concise and better stuctured

Reviewer #2: The results for the three groups, MB and PB patients and OD are not presented well, there is no clear cut comparison of the sensitivity and specificity between the MB, PB patients and the OD group. There should also be an evaluation shown of all six extraction buffers with a known amount of M. leprae whole cells from nude mouse footpad with the Ct scores shown to show which extraction solution is best.

Reviewer #3: the results are too detailed with regards to the intermediate steps before the final protocol. there should be the performances with regard to the test used in Brazil with the portable machine and the quantstudio on the clinical samples. the authors choose to have two targets, the 16S rDNA and the RLEP repeated element present only on Mycobacterium leprae genome. SInce this element is repeated 35 to 37 in the genomes, this PCR should be more sensitive with a shorter CT. this is well shown in the analytical sensitivity results. However, can the authors explain why the CT values are similar, or even shorter for the 16S results? (table 3).

Table 4: what is the gold standard for measuring the results of performances of the tests on the clinical specimens? clinical diagnosis? the PPV AND NPV is interesting only if we know the prevalence of the disease in the population included in the study. Because it was enriched in leprosy patients, this is not informative.

the protocol designed for simplifying the extraction is interesting but not really applicable in the field since it still requires a biology laboratory. can the atuhors simplify also this part by giving only the final results with the best protocol. 

minor comments: 

- explain better what indeterminate results are, and what is the result given to the clinician then?

- the discordant results between Q3 plus and QS5, could be detailed with regard to the gold standard

**Conclusions**

-Are the conclusions supported by the data presented?

-Are the limitations of analysis clearly described?

-Do the authors discuss how these data can be helpful to advance our understanding of the topic under study?

-Is public health relevance addressed?

Reviewer #1: -Are the conclusions supported by the data presented?yes but can be more structured

-Are the limitations of analysis clearly described? no limitations stated

-Do the authors discuss how these data can be helpful to advance our understanding of the topic under study?

-Is public health relevance addressed?yes

Reviewer #2: It is not possible the way the data were shown to evaluate if the conclusions are supported by the data because they should separate and show the Ct values for the MB, PB and OD groups. This would make evaluation of the usefulness of this test clearer.

Reviewer #3: the conclusions about a portable test for leprosy diagnosis are not reached yet. The protocol is still quite complicated 

(see figure 6) for a primary care center. We do not know the Q3-plus apparatus (need to describe shortly its advantages and disadvantages), is it easily found in the international market?

the authors could present only their final protocol with the analytical results and performances on the clinical samples.

need to explain what brings to the results the analyses of the 16S and the 18S rDNA, which is far too complicated for the field.

**Editorial and Data Presentation Modifications?**

Reviewer #1: (No Response)

Reviewer #2: This work reports the optimization of a molecular diagnostic point-of-care (POC) kit for detecting M. leprae in DNA extracted from leprosy patient skin lesions to establish thresholds, cutoffs and limits of detection (LOD) using 16S rRNA and RLEP targets while using synthetic DNA, purified DNA from M. leprae and pre-characterized clinical samples. Using the established complete protocol for simplified extraction and qPCR on the portable platform the LOD was established at 113.3 genome-equivalents/ul for 16S rRNA and 17.7 genome-equivalents/ul for RLEP with a sensitivity of 52% in reactions. The following comments and questions should be addressed.

1. Table 2 shows the six lysis solutions that were evaluated in the development of a simplified extraction method. Eventually, the optimal extraction method identified was solution 2 (line 290) composed of PBS (366 ul), proteinase K (14.6 ul), urea (146 ul) and nuclease free water (73.4 ul). What would have been ideal would be to show a comparison of the qPCR curves for a known quantity of M. leprae cells extracted with these six solutions to show the differences between detecting each of the M. leprae targets, 16S rRNA and RLEP. This could have been shown as LOD values obtained from these six extraction solutions. Table 5 shows something along these lines by evaluating the six solutions with the 18S rRNA as the target by qPCR in porcine skin samples but it would have been best to show the two M. leprae targets and their Ct scores. 

2. It was mentioned in the Discussion in lines 686-694 noted that as is the case with previous PCR and qPCR detection of various targets and biological materials, PB cases show a variable range of sensitivity from 36.4% to 85%. Lines 221-227 detail the numbers of MB and PB patients and other dermatoses (OD) where skin biopsies were extracted, also shown in supplementary Table S1. It is not clear from the Results section in lines 540-555 showing the numbers of samples tested (n = 53) what the positive “agent” is (16S rRNA or RLEP, either or both). The number seems to be an aggregate here but it does not differentiate between MB, PB and OD samples. Ideally, the number of positives should be reported for each separate group for each target since the percentage of positives would likely be higher for MB than PB. Since the BI for the majority of PB cases is 0, it would not be surprising to see that the sensitivity for the qPCR reaction using the NAT Hans kit on the Q3-Plus equipment would be low but this information is also important. Reporting the differences in the sensitivity between these two groups, MB versus PB, would be preferred. It is assumed almost all of the OD cases would be negative for both targets but it is not clear from the presentation of the data if this is the case.

3. The BI for MB cases in Table S1 ranges from 1-5.75. It is likely and expected that the majority of positive responses for the qPCR test would be positive for MB patients with a BI >2.0 but it would be good to mention this in the Discussion.

4. Two of the OD individuals in Table S1 have a positive BI, line 84 (1.5) and line 102 (4.5). Is this correct and were either of these individuals positive by qPCR? 

Minor points

Line 163, “mammals” should probably be “humans”

Table 2, unclear if the abbreviation PK stands for proteinase K?

Line 368, “a.u.” abbreviation not defined

Line 548, “testes” should be “tests”

Line 565, “sensibility” should be “sensitivity”

Line 574, “adjunctive” should be “adjunct”

Reviewer #3: delete many tables and figures or put them as supplementary data.

**Summary and General Comments**

Reviewer #1: Minor revision

A study that is needed, because we need more field friendly DNA extraction and qPCR for leprosy diagnosis. Overall the article woudl benefit from a better flow . maybe first evaluating the extraction and then the qPCR, now it is quite confusing which samples are used for what and what is analysed for what. Maybe a flowchart or outline of the steps undertaken that lead to the DNA ecxtractuion evaluation and the qPCR evaluation. I think the article can be more concise and better structured.

line 55: FI is not worldwide use abbreviation: The common abbrevation is GD G0D: grade-0 disability. G1D: grade-1 disability. G2D: grade-2 disability

line78: analytical sensitivity: the analytical sensitivity stays the same, the diagnostic sensitivity changes

line 81: abbreviate explanation of NAT and not for first time in line 102

lline 134-135; copies 16S or RLEP or 18S or all three?

line 144: double point

line 145: is this standard curve use to make the standard curve for the qPCR? or for evaluating the extraction? for the extraction it is the right choice, for the qPCR not, therefore plasmid DNA or gDNA should be used

table 1: what is the Gain? of what?

table 2: not clear in the table from where the skin biopsies come,human of pork? in line 215 3mm ksin biopsies are used , so what is th rationale for the 3mm skin biopsy?

Reviewer #2: See above comments.

Reviewer #3: the paper is far too long and too descriptive of intermediate steps to reach the final version of the protocol. The authors should focus on the performances on the clinical samples detailing the gold standard they choose to interpret their results.

PLOS authors have the option to publish the peer review history of their article (what does this mean?). If published, this will include your full peer review and any attached files.

Reviewer #1: No

Reviewer #2: Yes: John S. Spencer

Reviewer #3: No
---

## [Decision Letter · Decision Letter 1]

23 Sep 2024

Dear Dr. Costa,

We are pleased to inform you that your manuscript 'Validation of the performance of a point of care molecular test for leprosy: from a simplified DNA extraction protocol to a portable qPCR' has been provisionally accepted for publication in PLOS Neglected Tropical Diseases.

Best regards,

Katharina Röltgen

Academic Editor

Mathieu Picardeau

Section Editor

Reviewer's Responses to Questions

**Key Review Criteria Required for Acceptance?**

**Methods**

-Are the objectives of the study clearly articulated with a clear testable hypothesis stated?

-Is the study design appropriate to address the stated objectives?

-Is the population clearly described and appropriate for the hypothesis being tested?

-Is the sample size sufficient to ensure adequate power to address the hypothesis being tested?

-Were correct statistical analysis used to support conclusions?

-Are there concerns about ethical or regulatory requirements being met?

Reviewer #2: This is a revised version.

Objectives of study are clearly articulated with a testable hypothesis.

Study design is appropriate.

Study population is clearly described.

Sample size is sufficient.

Statistical analyses correctly used.

No ethical concerns.

**Results**

-Does the analysis presented match the analysis plan?

-Are the results clearly and completely presented?

-Are the figures (Tables, Images) of sufficient quality for clarity?

Reviewer #2: Analysis matches analysis plan.

Results are more clearly and completely presented.

Figures and Tables are of good quality.

**Conclusions**

-Are the conclusions supported by the data presented?

-Are the limitations of analysis clearly described?

-Do the authors discuss how these data can be helpful to advance our understanding of the topic under study?

-Is public health relevance addressed?

Reviewer #2: Conclusions are supported by the data.

Limitations of the analysis were described.

Authors adequately discuss how these data can advance our understanding of the topic.

Public health relevance is discussed.

**Editorial and Data Presentation Modifications?**

Reviewer #2: I am satisfied with the revisions to the previous manuscript.

**Summary and General Comments**

Reviewer #2: No further comments.

PLOS authors have the option to publish the peer review history of their article (what does this mean?). If published, this will include your full peer review and any attached files.

Reviewer #2: **Yes: **John S. Spencer

---

## [Editor Report · Acceptance letter]

2 Oct 2024

Dear Dr. Costa,

We are delighted to inform you that your manuscript, "Validation of the performance of a point of care molecular test for leprosy: from a simplified DNA extraction protocol to a portable qPCR," has been formally accepted for publication in PLOS Neglected Tropical Diseases.

Best regards,

Shaden Kamhawi

co-Editor-in-Chief

Paul Brindley

co-Editor-in-Chief
